# Pyrocatalysis—The DCF assay as a pH-robust tool to determine the oxidation capability of thermally excited pyroelectric powders

**Sascha Raufeisen**[1,2], **Michael Stelter**[1,2,3], **Patrick Braeutigam**[1,2]*

**1** Institute of Technical Chemistry and Environmental Chemistry, Faculty of Chemistry and Earth Sciences, Friedrich Schiller University Jena, Jena, Germany, **2** Center for Energy and Environmental Chemistry (CEEC Jena), Faculty of Chemistry and Earth Sciences, Friedrich Schiller University Jena, Jena, Germany, **3** Fraunhofer IKTS, Fraunhofer Institute for Ceramic Technologies and Systems, Hermsdorf, Germany

* patrick.braeutigam@uni-jena.de

**Data Availability Statement:** All relevant data are within the manuscript and its Supporting Information files.

## Abstract

Pyrocatalysis uses thermally excited pyroelectric materials for the generation of reactive oxygen species in water. This unique feature allows it to harvest energy in the form of natural temperature gradients or waste heat from industrial processes in order to degrade organic pollutants at low costs. Its further development into an advanced oxidation process for water remediation is dependent on the availability of pH-robust and nonspecific redox assays for the determination of its oxidation capability. Nevertheless, previous studies neglected the influence of pH changes and they were focused mainly on the degradation of one organic compound or specific chemical dosimetries. In this study, a pH-robust and nonspecific reaction protocol of the dichlorofluorescein assay was established for the investigation of the oxidation capability of the pyrocatalytic process. This reaction protocol was tested on three pyroelectric powders ($LiNbO_3$, $LiTaO_3$, $BaTiO_3$) in different amounts and it overcomes major constraints of a previously used dichlorodihydrofluorescein diacetate-based reaction protocol. Instead of its diacetate, dichlorodihydrofluorescein was used as fluorogenic probe and its concentration was drastically reduced to 1 µM. For the first time, these changes enable the determination and comparison of the oxidation capability independently of pH-rising processes, which are present for all investigated pyroelectric powders up to a pH of 11. Additionally, the precision of the dichlorofluorescein assay was drastically increased and the determination and consideration of autoxidation processes was enabled. Of all three pyroelectric powders, $BaTiO_3$ exhibited the highest oxidation capability with a linear increase with respect to the powder amount.

## Introduction

The number of potentially toxic synthetic and natural organic compounds contaminating fresh water resources at trace levels (ng/L—µg/L) has risen drastically [1]. The origin of these substances like pharmaceuticals, pesticides, industrial chemicals or flame-retardants is their uncontrolled discharge in the environment. A variety of pharmaceuticals, substances used in

**Funding:** The authors received no specific funding for this work.

**Competing interests:** The authors have declared that no competing interests exist.

the household and estrogens are emitted mainly unchanged or as metabolites/transformation products from urban wastewater treatment plants [2,3]. In order to avoid negative effects on aquatic ecosystems or human health, it is necessary to introduce an additional treatment stage to wastewater treatment plants.

One group of techniques that are capable of degrading a wide range of micropollutants in such a treatment stage are advanced oxidation processes (AOPs) [4]. AOPs are based mainly on the *in situ* generation of highly reactive oxygen species (ROS) which react rapidly and unselectively with organic molecules in water under formation of less toxic degradation products and $CO_2$ [4–7]. ROS such as OH can be generated by a variety of different techniques such as ozonation ($O_3$), photocatalysis (UV/$TiO_2$), Fenton ($Fe^{2+}$/$H_2O_2$), sonolysis and electrocatalysis [8–10]. All of these techniques reveal advantages and disadvantages regarding the use of toxic reagents and/or corrosive reagents ($O_3$, $H_2O_2$), the range of degradable micropollutants (polarity), the formation of toxic transformation products and the energy demand and costs. Especially the reduction of the energy demand for micropollutant removal is a major driving force for the development of new and the enhancement or combination of existing AOP techniques. One approach, the harvesting of energy from the environment, can be utilized only by solar photocatalysis owing to the development of new visible-light-absorbing photocatalysts [11].

A new candidate for a technique, which is able to cover a significant amount of its energy demand by energy harvesting, is an AOP technique called pyrocatalysis. The pyrocatalysis was introduced in 2012 by Gutmann et al. and uses thermally excited pyroelectric materials for water remediation [12]. This technique is based on the pyroelectric effect of these materials, which causes transient variations of their polarization magnitude under thermal cycling. While polarization charges on the materials surface are screened by compensation charges under equilibrium conditions, a transient net charge occurs during thermal excitation. These surface charges can grow high enough to trigger redox reactions, which generate ROS and enable the degradation of organic water pollutants. In this way, this technique has the potential to harvest energy in the form of natural temperature gradients or waste heat from industrial processes in order to reach a drastic reduction of the energy consumption as well as the costs for wastewater remediation [13].

Experimental studies of the pyrocatalysis have shown that several organic compounds (e.g. 2′,7′-dichlorodihydrofluorescein DCHF, coumarin, terephthalic acid, rhodamin B) can be oxidized in the presence of different thermally excited pyroelectric materials like $BaTiO_3$, $LiNbO_3$, $BiFeO_3$ or $NaNbO_3$ (Table 1). It was also possible to disinfect a solution of bacteria and to generate hydrogen with a pyrocatalytic technique [12,14]. The generation of reactive hydroxyl radicals OH and superoxide radicals $O_2^-$ during thermal excitation of pyroelectric materials was confirmed via electron spin resonance (ESR) spectroscopy, different chemical dosimetries and radical scavenger experiments (Table 1).

Beside these fundamental findings regarding the existence of a pyrocatalytic process, only little is known about the influence of critical reaction (e.g. pH, conductivity) and process parameters (e.g. heating/cooling rates, temperature range) or the exact mechanism of this process [29]. Moreover, the knowledge about the influence of important material parameters like the particle size/shape/porosity/surface, the pyroelectric constant or the crystallographic phase on the pyrocatalytic oxidation process is still vague. This knowledge is essential in order to design and synthesize effective pyroelectric catalysts and to optimize reaction as well as process parameters.

As shown in Table 1, only a few model contaminants with similar degradation properties or chemical dosimetries that are susceptible to only one type of ROS were used for investigation and optimization of the pyrocatalytic process in previous studies. This approach has the risk to optimize the material, reaction and process parameters towards this one model contaminant or type of ROS. The consequence would be a water remediation technique that is limited to

**Table 1. Comparison of pyrocatalytic studies regarding pyrocatalysts, degraded pollutants, types of dosimetries and radical scavenger.**

| pyrocatalyst | pollutant | dosimetry | radical scavenger | Ref. |
|---|---|---|---|---|
| $LiNbO_3$, $LiTaO_3$ | - | DCF | - | [12] |
| $Pd@BaTiO_3$ | - | CU | BMPO (ESR) | [15] |
| $BiFeO_3$ | RhB, (MO, MB) | TA | EDTA, BQ, TBA | [16] |
| $BaTiO_3$ | RhB | TA | - | [17] |
| ZnO | RhB | TA | EDTA, BQ | [18] |
| $NaNbO_3$ | RhB, (MB) | - | EDTA, BQ, IPA | [19] |
| $Ba_{0.7}Sr_{0.3}TiO_3$ | RhB | TA | EDTA, BQ, TBA | [20] |
| $BaTiO_3$ | RhB | - | EDTA, BQ, TBA | [21] |
| 2D black P | RhB | - | - | [22] |
| $NaNbO_3$ | RhB | TA | EDTA, BQ, IPA | [23] |
| $BaTiO_3$ | RhB | - | EDTA, BQ, TBA, $N_2$ | [24] |
| $BaTiO_3$ | RhB | - | EDTA, BQ, TBA | [25] |
| $ZnO@BaTiO_3$ | RhB, (MO, MB) | - | EDTA, BQ, TBA | [26] |
| $metal@BaTiO_3$ | RhB | TA | EDTA, BQ, TBA | [27] |
| $Pb(Zr_{0.52}Ti_{0.48})O_3$ | RhB | TA | - | [28] |

DCF: dichlorofluorescein, CU: coumarin, BMPO: 5-*tert*-butoxycarbonyl-5-methyl-1-pyrroline *n*-oxide, ESR: electron spin resonance, RhB: rhodamine B, MO: methyl orange, MB: methyl blue, TA: terephthalic acid, EDTA: disodium ethylenediaminetetraacetate, BQ: benzoquinone, TBA: *tert*-butyl alcohol, IPA: *iso*-propyl alcohol.

the degradation of e.g. organic dyes with similar polarities. Additionally, the influence of pH changes on the model contaminants, chemical dosimetries and the pyrocatalytic process itself was neglected so far. This approach can lead to under- or overestimations of the pyrocatalytic degradation efficiency. Previous studies have clearly shown that the degradation efficiency of AOPs is strongly dependent on pH [30,31]. Moreover, it was shown that different organic dyes can have different optimal pH for their photocatalytic degradation [32]. Therefore, the goal of our study was to find a pH-independent and universal method for indirect ROS detection with the aim to optimize the unselective overall oxidation capability of the pyrocatalysis. In this way, the pyrocatalytic process will be applicable for the oxidative removal of a broad variety of contaminants in a broad variety of matrices.

As a result, a reaction protocol of the 2',7'-dichlorofluorescein (DCF) redox assay for the investigation of the oxidation capability of thermally excited pyroelectric powders was established. The DCF assay is based on the oxidation of the non-fluorescent dye precursor DCHF, which is susceptible to a variety of ROS, into the highly fluorescent dye DCF (Fig 1) [33,34]. This reaction protocol was proven pH-robust and tested on three thermally excited pyroelectric powders $LiNbO_3$, $LiTaO_3$ and $BaTiO_3$ in different powder amounts. It overcomes the constraints of a previously used DCHF diacetate (DCFH-DA)-based reaction protocol. We propose that this novel reaction protocol of the DCF redox assay should be used in future pyrocatalytic studies as a universal tool for the optimization of reaction as well as process parameters and for the evaluation of new pyroelectric catalysts in a comparable manner. With the help of this tool, the pyrocatalysis can take the next step towards an energy harvesting water remediation technique with low energy costs.

## Material and methods

### 2.1 Reagents and chemicals

All chemicals except $LiNbO_3$ were used as received without any further purification. Powders of $LiNbO_3$ (LN, > 50 μm), $LiTaO_3$ (LT, ~200 mesh) and tetragonal $BaTiO_3$ (BT, 200 nm) were

**Fig 1. Alkaline hydrolysis of 2′,7′-dichlorodihydrofluorescein diacetate (DCHF-DA) into the deacetylated DCHF and further (aut)oxidation to 2′,7′-dichlorofluorescein (DCF).**

supplied by Alfa Aesar (LN, LT) and US Research Nanomaterials, Inc. (BT) with a purity of 99.9% (trace metal basis). The microcrystalline LN powder (4.7 g) was further milled in a planetary ball mill (Pulverisette P7 classic line, Fritsch GmbH) with a $ZrO_2$ beaker ($V$ = 45 mL) and $ZrO_2$ balls ($n$ = 18, $d$ = 10 mm) with 200 rpm for 3 h (10 min break every 30 min).

DCHF-DA ($>$ 97%), LiOH ($>$ 98%) and lithium acetate (LiOAc, reagent grade) were purchased from Sigma Aldrich. DCF (reagent grade) was purchased from Alfa Aesar. $BaCl_2 \cdot 2H_2O$ ($>$ 99%) and $NaH_2PO_4 \cdot 2H_2O$ ($>$ 98%) were supplied by Merck. NaOH ($>$ 98%), methanol ($>$ 99.8%) and pH buffer standards (pH 1.679/ 4.005/ 6.865/ 9.180 / 12.454 @ 25 ˚C traceable to N.I.S.T.) were purchased from VWR Chemicals. Ultrapure water (0.055 µS/cm, GenPure UV, Fisher Scientific GmbH) was used for sample preparation and analysis. All Stock solutions were stored in a refrigerator at 9 ˚C and were protected from light. DCHF-DA was stored in a refrigerator at -10 ˚C.

## 2.2 Analysis methods

pH values were measured after calibration (-58.2 mV/pH, 25 ˚C) of the pH electrode (phenomenal MIC220, Mikro) with a pH meter (MU 6100 L) and pH buffer standards from VWR. Fluorescence measurements of DCF were conducted with a fluorescence detector (FP-4025, Jasco) which was equipped with a square cell holder for 10 x 10 mm square cells (Type 3/G/10, Starna Scientific). The wavelengths for excitation and emission were 480 and 525 nm and a detector gain of 1 was used. In a typical measurement 800 µL of a sample were diluted with 2500 µL of a degassed 0.16 mM NaOH solution. In order to get reliable results it was crucial to protect the samples from light during the whole measurement process to prevent further DCHF oxidation. The fluorescence intensities correlated with the DCF concentration $c_{DCF}$ in a linear fashion (S1 Fig). The quantitative analysis was done by external calibration with standards in three concentration ranges and duplicate measurements. In S1 Table, calibration parameters for the linear regressions are displayed.

## 2.3 Powder characterization

The crystal structures of samples were characterized by using X-ray diffraction (XRD, Bruker Phaser D2) with Cu-K$\alpha$ radiation ($\lambda$ = 1.54056 Å) over the range of $2\theta$ from 10 to 80˚ with a scanning rate of 0.02˚/s. In order to obtain the unit cell parameters and weight fractions, a Rietveld refinement of the XRD data was performed. The instrumental broadening and shapes

of reflection profiles were calibrated and fitted with program MAUD [35] using the diffraction pattern of $LaB_6$ standard powder.

## 2.4 Experimental procedures

To compare the ROS-induced oxidation capability of three different pyroelectric powders in aqueous solution the DCF assay was used [36]. The initial DCHF-DA-based reaction protocol was only slightly modified in comparison to the method of Gutmann et al. [12]. DCHF-DA was dissolved in methanol (300 μL per mg DCHF-DA) and subsequently diluted in 25 mM $NaH_2PO_4$ (3.4 mL per mg DCHF-DA). Owing to the low solubility of DCHF-DA in water, partial precipitation took place and a fine DCHF-DA suspension was obtained. The DCHF-DA suspension was then diluted with ultrapure water to a final theoretical concentration of 96 μM.

The final DCHF-based reaction protocol is based on the method of Carthcart et al. which was further modified [33]. DCHF-DA was dissolved in methanol (160 μL per mg DCHF-DA) and 0.01 M NaOH (650 μL per mg DCHF-DA) was added to trigger a complete deesterification of DCHF-DA to DCHF within 30 min. Finally, 25 mM $NaH_2PO_4$ (1.8 mL per mg DCHF-DA) was used to neutralize the base and the DCHF solution was diluted with ultrapure water to a concentration of 2 μM. In order to get reliable results all solvents were degassed prior to use and all stock, reaction and sample solutions were protected from light and air during all procedures.

In a typical experiment the pyroelectric powder (2.9–116.7 mg) was placed within a micro tube (PP, amber, $V_{max}$ = 1.85 mL) and suspended in 875 μL water, a base or salt solution. Afterwards, 875 μL of the DCHF-DA or DCHF reaction solution were added, the micro tube was sealed with Parafilm and shaken thoroughly. The thermal excitation of the suspended pyroelectric powders was realized with the help of a thermomixer (heating/cooling thermo-mixer MKR13, Hettich Benelux) equipped with an aluminium block for 24 micro tubes. 12 micro tubes were placed in every second slot of the aluminium block while the remaining slots were left free (S2 Fig). The oxidation experiments were carried out with temperature cycles between 34.5 and 64.0 ˚C inside the micro tubes and simultaneous shaking at 1200 rpm. The internal temperature $\Theta_{in}$ was measured with a type K thermocouple (NiCr-Ni, ± 1.5 ˚C) directly placed into the reaction tube. Prior to the temperature program, the reaction tubes were equilibrated for 5 min in the pre-heated aluminium block (32.5 ˚C). The temperature program consisted of nine cycles with 18 min length (Fig 2) and it was followed by a 5 min cooling phase back to 20 ˚C (S3 Fig). The preset temperature $\Theta_P$ and the measured temperature of the aluminium block $\Theta_{Al}$ (internal thermocouple) are displayed in S4 Fig.

The heating and cooling rates reached values up to +4.6 K/min or -4.9 K/min with an average of +3.25 or -3.25 K/min, respectively (Fig 2). Finally, the samples were centrifuged twice at 20 ˚C (17300 rcf, 5 and 15 min) to remove all particles and the fluorescence intensity of the supernatant was measured immediately. Every experiment was performed twice and the relative error of $c_{DCF}$ corresponds to the standard deviation.

## Results and discussion

### 3.1 Powder characterization

The phase compositions of pyroelectric powders were analyzed by XRD, as shown in S5–S7 Figs and S2 Table. For BT the XRD pattern indicates a mixture of paraelectric cubic (COD 1507757, $Pm\bar{3}m$) and ferroelectric tetragonal phase (COD 1507756, $P4mm$) as the peak at $2\theta$ of 45˚ shows a strong but not complete splitting (S5 Fig inset) [37,38]. The phase composition

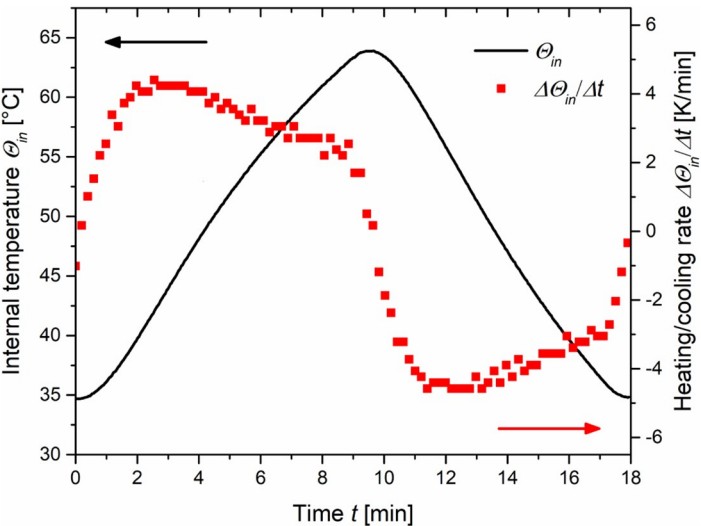

**Fig 2. Measured temperature inside the reaction vessel $\Theta_{in}$ over time $t$ and corresponding heating/cooling rate $\Delta\Theta_{in}/\Delta t$ for one temperature cycle used in the DCHF-oxidation experiments.**

of BT was further investigated through diffraction Rietveld refinement method and the results are listed in S2 Table. According to the XRD Rietveld refinement results, the weight fractions of ferroelectric tetragonal and paraelectric cubic $BaTiO_3$ phases are 75.0% and 25.0% and the lattice parameters are $a_{tetr}$ = 3.995 Å, $c_{tetr}$ = 4.033 Å and $a_{cubic}$ = 4.009 Å respectively. For LN (COD 1541936, *R3c*) no impurity peaks could be observed whereas the XRD pattern of LT (COD 2101846, *R3c*) exhibits $Ta_2O_5$ (COD 1531068, *Pccm*) as impurity phase (S6 and S7 Figs) [39–41]. Rietveld refinement yielded a weight fraction for the $Ta_2O_5$ impurity of 10.9%.

### 3.2 DCHF-DA-based DCF assay (Gutmann et al.)

As a starting point for the comparison of the overall oxidation capability of three different pyroelectric powders LN, LT and BT a reaction protocol based on the work of Gutmann et al. was used [12]. In this protocol, DCHF-DA is used directly and the pH-dependent deesterification into the oxidation-prone DCHF has to be initiated in situ (Fig 1). In contact with water, the LN and LT powders Gutmann et al. used underwent leaching processes, which significantly increased the pH of the reaction solution. One research question of this study was to find out whether this DCHF-DA-based reaction protocol can have a universal applicability regarding the powder type and powder amount. It seems likely that pyroelectric powders, which do not underlie intense leaching processes in water, are not able to convert enough DCHF-DA into DCHF. The same applies to lower powder amounts, which could not be able to rise the pH above a certain level. In these cases, the resulting $c_{DCF}$ should not be (only) dependent on the type and amount of pyroelectric powder but also on the increased pH as a consequence of leaching processes.

In order to investigate whether there is a dependence of $c_{DCF}$ from the pH, the pH values of the DCHF-DA reaction solutions after application of the temperature program were measured (Fig 3a). This was done for three pyroelectric powders (LN, LT, BT) and different powder amounts $\beta$ between 0.7 and 66.7 mg/mL.

Fig 3a demonstrates the great differences of the pH for different pyroelectric powders regarding $\beta$. Without powder ($\beta$ = 0 mg/mL), the pH was 5.3 due to $NaH_2PO_4$ in the

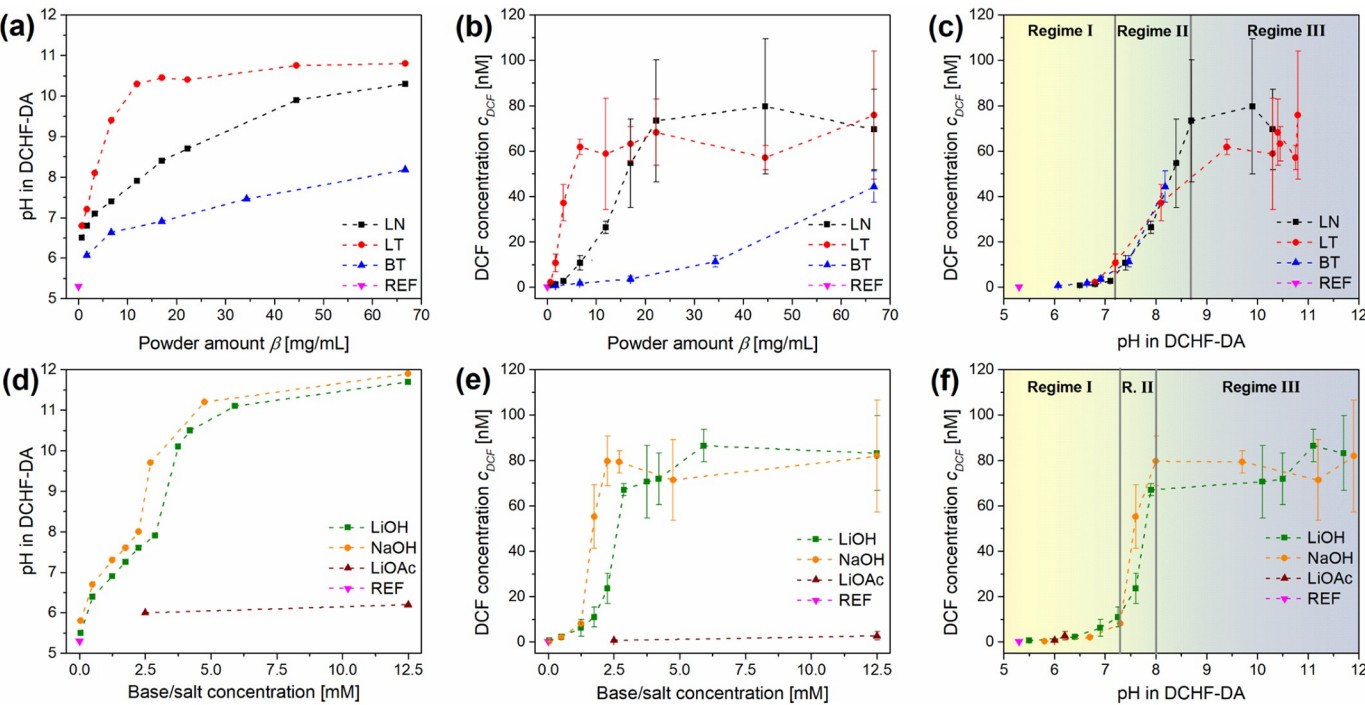

**Fig 3. Results with the DCHF-DA-based reaction protocol of the DCF assay for different pyroelectric powders, bases/salts and reference experiments (REF) after thermal treatment.** pH-increase with increasing powder amount $\beta$ (a) or base/salt concentration (d). Measured DCF concentration $c_{DCF}$ with increasing $\beta$ (b) or base/salt concentration (e). Combination of measured pH and $c_{DCF}$ for powders (c) and bases/salts (f). LN: LiNbO$_3$, LT: LiTaO$_3$, BT: BaTiO$_3$, DCF: dichlorofluorescein, DCHF-DA: dichlorodihydrofluorescein diacetate.

DCHF-DA reaction solution. For LT a drastic rise of the pH above 10 ($\beta_{LT}$ = 11.7 mg/mL) was observed which weakened until it reached 10.8 for the highest $\beta$. For LN and BT, the pH values rose slower and lower maximum pH values of 10.3 and 8.2 were reached respectively. The findings for LN and LT coincide with the findings of Gutmann et al. [12]. They found an increase of the pH for different nano-sized LN and LT powders above 10 at comparable values for $\beta$. The mechanism behind this phenomenon was found to be an ion exchange between H$^+$ and Li$^+$ whereby H$_x$Li$_{1-x}$NbO$_3$ is formed [42,43]. For BT the dissolution of Ba$^{2+}$ and pH shifts from aqueous solutions of nano-sized powders were studied and described by Tripathy and Raichur [44]. Although they only stirred BT suspensions at room temperature, they found high pH shifts depending on the initial pH. For suspensions with $\beta_{BT}$ of 250 mg/mL and initial pH values of 4.5 and 7.0 the pH rose in 3 h up to 6.3 and 7.7 respectively.

Due to the drastic increase of the pH and the possible impact on the DCF molecule, fluorescence spectra of DCF at different pH values were measured (S8 Fig). It was found that the relative fluorescence intensity increased about 10% when the DCF solution (150 nM) was diluted with 0.1 mM NaOH at a pH of 9.4 instead of water (pH 5.3). This pH-dependence of DCF fluorescence is in good agreement with literature findings [45]. With the aim of excluding any pH-dependence on the analysis results, all samples were diluted with 0.16 mM NaOH before the fluorescence measurements. This procedure enables the comparison of samples with lower pH (e.g. reference experiments) and samples with higher pH.

After the pH-robustness of the fluorescence measurement of DCF was ensured, the oxidation capability of the pyroelectric powders was investigated. For this purpose, the amount of oxidized DCHF into the fluorescent DCF $c_{DCF}$ was determined as a function of $\beta$ (Fig 3b). Like

for the increase of the pH values, LT showed a drastic increase of $c_{DCF}$ even for low $\beta_{LT}$. At a $\beta_{LT}$ of 6.7 mg/mL, $c_{DCF}$ reached a plateau at 64 nM (± 16 nM) without any further significant increase. For LN, the increase of $c_{DCF}$ was less sharp and a plateau at 74 nM (± 22 nM) was reached at a $\beta_{LN}$ of 22.2 mg/mL. It seems that for LN and LT a maximum of $c_{DCF}$ exists which shows no significant difference for both powders. On the contrary, BT showed a steady increase of $c_{DCF}$ up to 44.3 nM at the highest $\beta_{BT}$ with a moderate standard deviation. For LN and LT the standard deviation of $c_{DCF}$ also seems to be dependent on $\beta$. As soon as $c_{DCF}$ approached the plateau value, the relative standard deviation increased drastically up to 42%. An explanation for these high standard deviations could be that DCHF-DA cannot completely dissolve in water whereby the DCHF amount per reaction vessel varies. Another explanation is the high DCHF-DA concentration in comparison to the resulting DCF concentration (~0,2%). For this reason, DCF amounts originated from autoxidation processes should have a high influence on the total DCF concentration.

In order to determine their interdependence, $c_{DCF}$ was plotted as a function of the pH (Fig 3c). It is important to notice that this plot of $c_{DCF}$(pH) shows no difference between all three powders up to a pH of 8.3. Up to a pH of 7.2, nearly no $c_{DCF}$ was observed. Starting at pH 7.2 the plot shows a sharp increase of $c_{DCF}$ until a plateau is reached at pH 8.7. This plateau is slightly but not significantly higher for LN than for LT. For this reason, it can be assumed that the increase of $c_{DCF}$ is dominated by the increase of the pH when the DCHF-DA-based reaction protocol is used. Most of this behavior can be explained with the degree of DCHF-DA deesterification. DCHF-DA has to be hydrolyzed into DCHF before the thermally excited pyroelectric powders are able to oxidize it into DCF (Fig 1). With this knowledge, Fig 3c can be divided in three pH regimes: Regime I without or with insufficient hydrolysis (pH 5.3 to 7.2), regime II with partial hydrolysis (pH 7.2 to 8.7) and regime III with sufficient hydrolysis of DCHF-DA (pH > 8.7).

In a next step, reference experiments were performed to verify the dominance of the pH on the DCHF-DA-based reaction protocol of the DCF assay. In these experiments, NaOH or LiOH instead of the pyroelectric powders were added to increase the pH of the DCHF-DA solution. Moreover, LiOH or LiOAc were added to investigate the effect of exchanged Li$^+$ with and without an increased pH. The rest of the experimental procedures were kept identical. In Fig 3d the pH increase as a consequence of the addition of different base or salt amounts is shown. Both bases showed a different behavior than the pyroelectric powders. The pH increased in two steps like in an acid-base titration and was slightly higher with NaOH. This behavior of the pH was expected due to the $NaH_2PO_4$ (4.2 mM) in the DCHF-DA solution, which acts as a buffer. When LiOAc was added, the pH increased to ~6 which lies between the pH of a pure solution of $NaH_2PO_4$ and a pure solution of LiOAc.

Fig 3e shows the amount of DCF, which was formed only due to the addition of different concentrations of bases or LiOAc without any pyroelectric powder. In the beginning $c_{DCF}$ increased only slowly for both bases until a concentration of 1.3 mM (NaOH) or 1.8 mM (LiOH) was reached. From this point on $c_{DCF}$ increased drastically until it reached a plateau at 78 nM ± 15 nM for NaOH and 76 nM ± 13 nM for LiOH. This behaviour was similar to LT. Nearly no DCF was formed when LiOAc was added. Like for LN and LT the standard deviation of $c_{DCF}$ increased drastically at higher DCF concentrations. These results were unexpected because they indicate that high amounts of DCHF were converted under thermal excitation to DCF even without the pyroelectric powders. This means that strong autoxidation processes have taken place as a result of residual dissolved oxygen in the reaction solution as well as unavoidable contact with air and light during the centrifugation and measurement procedures (Fig 1) [46].

Again, $c_{DCF}$ was plotted as a function of the pH (Fig 3f). This plot shows no difference between both bases. Up to a pH of 7.3, nearly no $c_{DCF}$ was observed. Starting at pH 7.3 the plot shows an increase of $c_{DCF}$, which is sharper than for the pyroelectric powders, until a plateau is

reached at pH 8. The plateau value of $c_{DCF}$ shows no significant difference between LiOH and NaOH. Like for the pyroelectric powders this plot demonstrates that the increase of $c_{DCF}$ is dominated by the increase of the pH when the DCHF-DA-bases reaction protocol is used. As a result, Fig 3f can also be divided in three regimes with different degrees of DCHF-DA deesterification. In comparison with the pyroelectric powders, the regime II with partial hydrolysis of DCHF-DA (pH ~7.3 to ~8) started slightly later and was narrower when the pH was increased with bases. However, the plateau values of $c_{DCF}$ in the regime III with sufficient hydrolysis show no significant differences for both bases, LN and LT. This means that the oxidation capability of thermally excited pyroelectric powders was not significantly higher than simple autoxidation processes. Consequently, the DCHF-DA-based reaction protocol has no universal applicability for a pH-independent investigation of the oxidation capability of different types and amounts of thermally excited pyroelectric powders.

### 3.3 DCHF-based DCF assay (Cathcart et al.)

The results obtained with the DCHF-DA-based reaction protocol of the DCF assay were the starting point for its modification. The main goal was the exclusion of the pH-dependence on the results of the reaction protocol. For this purpose, a reaction protocol of Carthcart et al. was adopted. In this protocol the pH-dependent step, the alkaline deesterification of DCHF-DA into the oxidation-prone DCHF, was performed ex situ prior to the experiment (Fig 1) [33]. The first experiments with this modified DCHF-based reaction protocol were done solely with BT. It exhibited the greatest pH-dependence with the DCHF-DA-based reaction protocol (Fig 3c). Therefore, it should show the greatest changes in the plot of $c_{DCF}$ as a function of pH.

It is shown in S9 Fig that the pH with this reaction protocol was only slightly higher in comparison with the DCHF-DA-based reaction protocol. This can be accounted to the addition of NaOH for the deesterification of DCHF-DA, which results in a lower concentration of $NaH_2PO_4$. The DCF concentration after thermal excitation of BT suspensions in the DCHF reaction solution can be seen in Fig 4a.

It was found out that a significant amount of DCHF (35 nM) was oxidized to DCF as a consequence of autoxidation processes even in the reference experiments without BT. This means that the measured $c_{DCF}$ is always a combination of the amount of DCF, which formed due to autoxidation processes during the experiments ($c_{DCF,A}$) and DCF amounts, which formed due to the thermal excitation of the pyroelectric powders or the addition of bases or salts ($c_{DCF,T}$). It was assumed that $c_{DCF,A}$ equals the DCF concentration of the reference experiment. Therefore, $c_{DCF,T}$ was extracted by subtracting $c_{DCF,A}$ from $c_{DCF}$ of experiments where pyroelectric powders, bases or salts were added (Fig 4a). In addition to the high value for $c_{DCF,A}$, the standard deviations for $c_{DCF}$ were high over the whole range of $\beta_{BT}$. Nevertheless, $c_{DCF}$ increased drastically in comparison with the DCHF-DA-based reaction protocol and a linear correlation between $c_{DCF,T}$ and $\beta_{BT}$ was found (1.89 nM DCF per mg/mL BT). Furthermore, $c_{DCF}$ seems to be less dependent from the pH with this DCHF-based reaction protocol as high DCF amounts were found even at low pH values below 8.7 and even 7.3 (Fig 4b). This means that the main goal of the modification of the reaction protocol was partially achieved. However, the possibility of a partial pH-dependence has to be excluded via suitable reference experiments. In addition, the fraction of autoxidation processes ($c_{DCF,A}$) was still too high and the precision of $c_{DCF}$ too low for this modified DCHF-based reaction protocol.

### 3.4 DCHF-based DCF assay (Raufeisen et al.)

With the aim of reducing $c_{DCF,A}$ (autoxidation) and the standard deviation of $c_{DCF}$ (precision), the DCHF-based reaction protocol was further modified. This was done by decreasing the

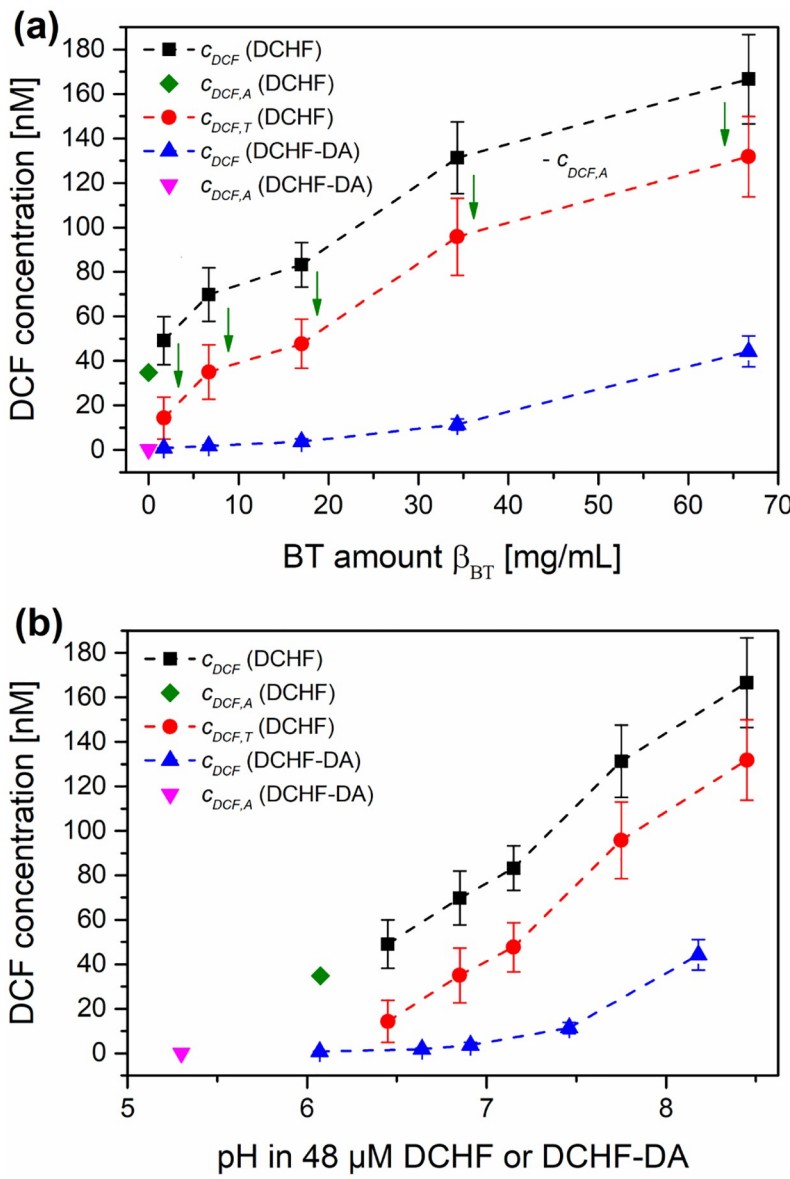

**Fig 4. Comparison of the results with the DCHF-DA- and DCHF-based reaction protocols ($c_0$ = 48 μM) of the DCF assay for BaTiO$_3$ powder (BT) and reference experiments (REF) after thermal treatment.** Measured DCF concentration with increasing BT amount $\beta_{BT}$ (a) and combination of measured pH and DCF concentration (b). Total DCF concentration $c_{DCF}$, DCF concentration formed due to autoxidation processes $c_{DCF,A}$ or due to thermal excitation of BT $c_{DCF,T}$, initial concentration $c_0$, DCF: dichlorofluorescein, DCHF: dichlorodihydrofluorescein, DA: diacetate.

initial DCHF concentration $c_{DCHF,0}$ drastically from 48 μM to 1 μM. A reduction of $c_{DCHF,0}$ is possible because less than 1% (167 nM) of the 48 μM DCHF were converted to DCF. The further modified reaction protocol was applied to LN, LT and BT (Fig 5a and 5c). As a result, $c_{DCF,A}$ decreased drastically from 35 nM down to 4 ± 1 nM. In addition, the standard deviation of $c_{DCF,T}$ for BT decreased drastically, especially for higher powder amounts (Fig 5b). Like for the DCHF-based reaction protocol after Cathcart et al. ($c_{DCHF,0}$ = 48 μM) a linear correlation between $c_{DCF,T}$ and $\beta_{BT}$ was found (1.19 nM DCF per mg/mL BT). Surprisingly the slope of the linear regression was not even halved although $c_{DCHF,0}$ was reduced by a factor of 48. Up to 79

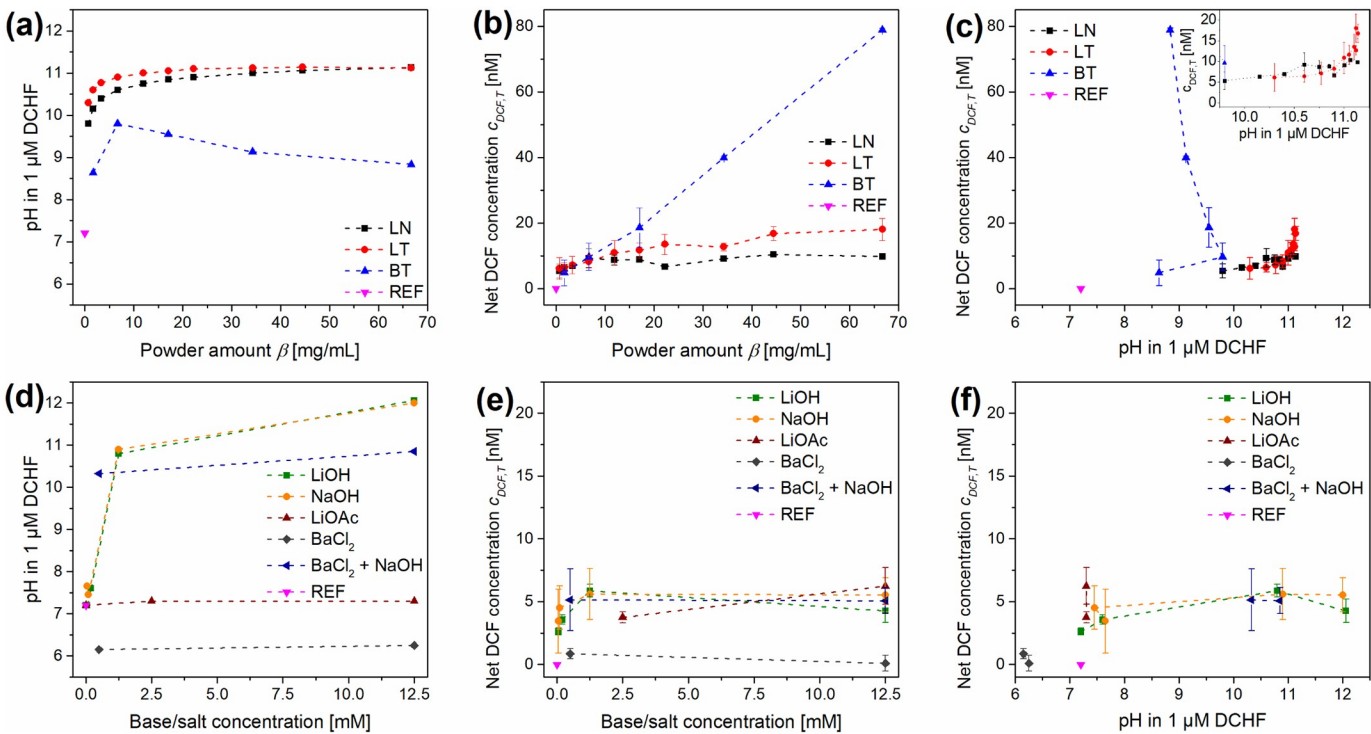

**Fig 5. Results with the DCHF-based reaction protocol ($c_{DCHF,0}$ = 1 μM) of the DCF assay for different pyroelectric powders, bases/salts and reference experiments (REF) after thermal treatment.** pH-increase with increasing powder amount $\beta$ (a) or base/salt concentration (d). Measured net DCF concentration $c_{DCF,T}$ with increasing $\beta$ (b) or base/salt concentration (e). Combination of measured pH and $c_{DCF,T}$ for powders (c) and bases/salts (f). LN: LiNbO$_3$, LT: LiTaO$_3$, BT: BaTiO$_3$, DCF: dichlorofluorescein, DCHF: dichlorodihydrofluorescein.

nM DCF were found at the highest $\beta_{BT}$. In contrary, the LN and LT powders showed an unexpected behavior. Despite $\beta_{LN}$ and $\beta_{LT}$ were increased by a factor of 95, $c_{DCF,T}$ rose only slightly by a factor of 1.7 (LN) or 2.9 (LT). In addition, LN and LT showed a much lower $c_{DCF,T}$ than BT especially at high powder amounts. Only up to 10 nM (LN) or 18 nM (LT) DCF were generated through thermal excitation of both pyroelectric powders. Furthermore, the offsets for the linear regressions of both powders were exceptionally high (7 nM vs. 1 nM for BT). From these results, it can be concluded that the greatest part of $c_{DCF}$, which was measured with the DCHF-DA-based reaction protocol, was generated through autoxidation processes. Only this modified DCHF-based reaction protocol enables an independent determination of $c_{DCF,A}$ of reference experiments and an extraction of $c_{DCF,T}$.

However, also for this reaction protocol the correlation between $c_{DCF,T}$ and the pH was investigated (Fig 5a and 5c). Without powder ($\beta$ = 0 mg/mL), the pH of the DCHF reaction solution was 7.2. The pH-increase in the case of LN and LT was much sharper than for the DCHF-DA-based reaction protocol. The reason is the much lower NaH$_2$PO$_4$ concentration, which results from the preparation procedure of the DCHF reaction solution with this reaction protocol (neutralisation with NaOH; high DCHF dilution). The pH for LN and LT increased up to 9.8 and 10.3 for the lowest powder amount and increased further up to 11.1 for the highest amount of both powders. The pH values for BT did not show an intuitive development. The pH values increased up to 9.8 for a $\beta_{BT}$ of 6.7 mg/mL and decreased from this point down to 8.8.

Again, $c_{DCF,T}$ was plotted as a function of the pH in Fig 5c. This plot shows that a higher $c_{DCF,T}$ was obtained for BT at much lower pH values in comparison to LN and LT. Consequently, this plot indicates that the increase of $c_{DCF,T}$ is not determined by the increase of the

pH as a consequence of leaching processes. This means that the oxidation of DCHF to DCF is only dependent on the type and amount of the thermally excited pyroelectric powder.

In order to verify these findings, several reference experiments were performed. Different bases (NaOH, LiOH), salts (LiOAc, $BaCl_2$) or combinations ($BaCl_2$ plus NaOH) were added to the DCHF reaction solution instead of the pyroelectric powders. LiOH, LiOAc, $BaCl_2$ and $BaCl_2$ plus NaOH were used to investigate the effect of exchanged $Li^+$ or $Ba^{2+}$ ions alone and in combination with an increased pH. The rest of the experimental procedures were kept identical. In Fig 5d the pH increase as a consequence of the addition of different base or salt concentrations is shown. As for LN and LT, the value increased drastically above 10 even for low amounts of the bases and reached values of 12 for the highest amounts of LiOH and NaOH. For LiOAc, $BaCl_2$ and $BaCl_2$ plus NaOH the pH values kept nearly independent of the salt concentrations at ~7.2, ~6.2 or ~10.6. Fig 5e shows the amount of DCF $c_{DCF,T}$, which was formed only due to the addition of different concentrations of bases and/or salts without any pyroelectric powder. For all bases and/or salts with the exception of $BaCl_2$ only between 2.6 and 6.5 nM DCF were found after thermal excitation. For these bases and/or salts $c_{DCF,T}$ was nearly independent of the investigated concentration range. In contrast, $c_{DCF,T}$ for the reference experiments with $BaCl_2$ was nearly 0 nM. From these results, it can be concluded that, as long as a base (pH > 10) or LiOAc is added, only a low amount of 4.5 ± 2 nM DCF will be generated during thermal excitation. The mechanism behind this matrix effect is unknown but this behaviour can partially explain the offset (~7 nM) of the linear regressions of LN and LT in Fig 5b. This offset is most likely a result of multiple matrix effects. When only $BaCl_2$ or BT (offset only 1 nM) is added, nearly no (additional) DCF will be formed. This is a hint that $Ba^+$ alone or in combination with an elevated pH < 10 will not cause any matrix effects, which rise $c_{DCF,T}$.

## Conclusions

In conclusion, a DCHF-based reaction protocol of the DCF assay for the investigation of the oxidation capability of different types and amounts of thermally excited pyroelectric powders was established. This reaction protocol was tested on three pyroelectric powders (LN, LT, BT) and it overcomes the constraints of the previously used DCFH-DA-based reaction protocol.

1. It enables the determination of the oxidation capability independently of pH changes which are present for nano-/microparticles between pH 7 and 12 (Fig 5a and 5c).

2. It exhibits a drastically increased precision (Fig 5b) and

3. it enables the determination and consideration of autoxidation processes (Fig 4a).

   The main modifications on the reaction protocol to meet these requirements were:

i. the use of a prehydrolyzed DCHF reaction solution via an *ex situ* alkaline deesterification of DCHF-DA (Fig 1) and

ii. the reduction of the initial DCHF concentration from 48 to 1 μM.

   Moreover, the addition of NaOH to the samples before the fluorescence measurements proved to increase the comparability between samples with different pH values. The reliability of all findings was proven by a range of proper reference experiments to exclude or quantify interfering effects of increased pH values or cation concentrations. We propose that this new pH-robust DCF redox assay should be used in future pyrocatalytic studies as a universal tool for a detailed investigation of material, reaction and process parameters. In this way, the unselective overall oxidation capability of the pyrocatalysis can be maximized in a comparable manner. This method is needed as previous studies dealing with the pyrocatalytic process were

focussed only on a few model contaminants with similar degradation properties or chemical dosimetries that are susceptible to only one type of ROS (Table 1). These studies also neglected the influence of pH changes. The risk of this approach is that the material, reaction and process parameters are optimized towards the oxidative removal of organic compound with similar polarities and to develop a water remediation technique, which is susceptible to pH changes. The DCF assay, on the other hand, is a pH-robust and universal tool, which allows the indirect detection of a variety of ROS in a broad variety of matrices. With this tool, the pyrocatalytic process can be developed into a new, energy harvesting AOP technique for water remediation with low energy costs.

## Supporting information

**S1 Fig. Results and linear regression of external calibration of DCF with fluorescence spectroscopy in three concentration ranges.** (a) 1–10 nM, (b) 10–150 nM, (c) 150–1000 nM.
(PDF)

**S2 Fig. Schematic sketch of the aluminium block of the thermomixer with 24 slots.** In a typical experiment, 12 micro tubes were placed in every second slot (numbered slots).
(PDF)

**S3 Fig. Measured temperature inside the reaction vessel $\Theta_{in}$ over time t and corresponding heating/cooling rate $\Delta\Theta_{in}/\Delta t$ for the whole temperature program used in the DCHF-oxidation experiments.** It consists of a 5 min equilibration phase at 32.5 ˚C followed by nine full cycles and a 5 min cooling phase back to the starting temperature.
(PDF)

**S4 Fig. Measured temperature inside the reaction vessel $\Theta_{in}$, of the aluminium block of the thermoshaker $\Theta_{Al}$ and the preset temperature $\Theta_P$ over time t for one temperature cycle used in the DCHF-oxidation experiments.**
(PDF)

**S5 Fig. XRD diffraction Rietveld refinement results of $BaTiO_3$ powder.** Inset shows composition of 002/020 reflection with respect to cubic and tetragonal fraction.
(PDF)

**S6 Fig. XRD diffraction Rietveld refinement results of $LiNbO_3$ powder.**
(PDF)

**S7 Fig. XRD diffraction Rietveld refinement results of $LiTaO_3$ powder with 11% $Ta_2O_5$ content.**
(PDF)

**S8 Fig. Fluorescence excitation and emission spectra of a DCF solution diluted with water or different concentrations of NaOH or HCl ($\lambda_{ex}$ = 480 nm; $\lambda_{em}$ = 525 nm) for pH adjustment.** Inset plot: Relative intensity of the DCF solution at different pH values.
(PDF)

**S9 Fig. Comparison of the pH-increase with the DCHF-DA- and DCHF-based reaction protocols ($c_0$ = 48 μM) of the DCF-assay for increasing amounts $\beta_{BT}$ of $BaTiO_3$ powder (BT) and reference experiments (REF) after thermal treatment.** DCHF: dichlorodihydrofluorescein, DA: diacetate.
(PDF)

**S1 Table. Calibration parameters of the linear regression for the quantification of DCF via fluorescence spectroscopy.**
(PDF)

**S2 Table. Crystallographic data of Rietveld refinement for BT, LN and LT powders.**
(PDF)

## Acknowledgments

The authors thank Johannes Buchheim for his help with the analysis of the X-ray diffractograms and Rietveld refinements.

## Author Contributions

**Conceptualization:** Sascha Raufeisen, Patrick Braeutigam.

**Data curation:** Sascha Raufeisen.

**Formal analysis:** Sascha Raufeisen.

**Investigation:** Sascha Raufeisen.

**Methodology:** Sascha Raufeisen.

**Project administration:** Patrick Braeutigam.

**Supervision:** Michael Stelter, Patrick Braeutigam.

**Validation:** Sascha Raufeisen.

**Visualization:** Sascha Raufeisen.

**Writing – original draft:** Sascha Raufeisen.

**Writing – review & editing:** Sascha Raufeisen, Michael Stelter, Patrick Braeutigam.

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
