## [Decision Letter · Decision Letter 0]

29 Nov 2019

PONE-D-19-27422

Pyrocatalysis - How to measure the oxidation capability of thermally excited pyroelectric powders

PLOS ONE

Dear Dr. Braeutigam,

Thank you for submitting your manuscript to PLOS ONE. After careful consideration, we feel that it has merit but does not fully meet PLOS ONE’s publication criteria as it currently stands. Therefore, we invite you to submit a revised version of the manuscript that addresses the points raised during the review process.

We would appreciate receiving your revised manuscript by Jan 13 2020 11:59PM. To enhance the reproducibility of your results, we recommend that if applicable you deposit your laboratory protocols in protocols.io, where a protocol can be assigned its own identifier (DOI) such that it can be cited independently in the future. For instructions see: http://journals.plos.org/plosone/s/submission-guidelines#loc-laboratory-protocols

We look forward to receiving your revised manuscript.

Kind regards,

Mason Sarafraz

Academic Editor

PLOS ONE

Journal Requirements:

Reviewers' comments:

Reviewer's Responses to Questions

**Comments to the Author**

1. Is the manuscript technically sound, and do the data support the conclusions?

Reviewer #1: Yes

Reviewer #2: Partly

2. Has the statistical analysis been performed appropriately and rigorously? 

Reviewer #1: Yes

Reviewer #2: N/A

3. Have the authors made all data underlying the findings in their manuscript fully available?

Reviewer #1: Yes

Reviewer #2: Yes

4. Is the manuscript presented in an intelligible fashion and written in standard English?

Reviewer #1: Yes

Reviewer #2: No

5. Review Comments to the Author

Reviewer #1: Overall, a well written manuscript. A few clarification/grammar/word chose comments.

1. Page 8, line 37. drop the "an"

2. Page 13, line 144. prior "to" use (add the "to")

3. Page 13, line 146. change "was given into" to "was placed within"

4. Page 14, line 191. change "drastically rose" to "significantly increased"

5. Page 20, line 320. This sentence seems to be missing words/text.

6. Page 21. line 355. change "these results can" to "these results it can"

7. PAge 22, line 395 change "these results can be" to "these results it can"

Reviewer #2: The correction points of this paper are listed below

a) The authors might choose more suitable title for their manuscript.

b) The novelty of this work isn’t clear. The authors should clarify the novelty of this work. It is better to show the novelty of this work by comparing it with the previous researches. Try to prepare a table for comparing the results.

c) Enhance the literature review of articles by using recent articles especially from journal of PLOS ONE. Most part of references are too old.

d) The abbreviation should be removed from ABSTRACT.

e) The language/writing needs improvement throughout the manuscript.

f) More explain should be presented in the result section.

a) My suggestion is that enhance the introduction section by splitting it to more paragraphs. Try to discuss the aim of the study and the novelty of the research in more details.

b) Abstract and conclusion must be improved.

c) All of figures don’t have good enough quality. The authors should correct them.

d) Keywords should be written!!

e) There is not any mention to the application of the results of this study in industry, present some application of this work.

f) The below paragraph needs some references

Page 3 line 69: “In previous studies, only a few model contaminants with similar degradation properties or 70 chemical dosimetries that are susceptible for only one type of ROS were used for investigation 71 and optimization of the pyrocatalytic process. This approach has the risk to optimize the 72 material, reaction and process parameters towards this one model contaminant or type of ROS. 73 Additionally, the influence of pH changes on the model contaminants, chemical dosimetries 74 and the pyrocatalytic process was neglected so far. Instead, we propose to use a pH-independent 75 and less specific redox assays for indirect ROS detection with the aim to optimize the 76 unselective overall oxidation capability of the pyrocatalysis. In this way, the pyrocatalytic 77 process will be applicable for the oxidative removal of a broad variety of contaminants in a 78 broad variety of matrices”

6. PLOS authors have the option to publish the peer review history of their article (what does this mean?). If published, this will include your full peer review and any attached files.

Reviewer #1: No

Reviewer #2: No

---

## [Author Response · Author response to Decision Letter 0]

18 Dec 2019

Dear Editor and Reviewers:

Thank you for your letter and comments concerning the manuscript entitled “Pyrocatalysis - How to measure the oxidation capability of thermally excited pyroelectric powders” (Manuscript number: PONE-D-19-27422). The comments are all valuable and very helpful for revising and improving our paper. We have totally revised the manuscript. Based on the comments we corrected the manuscript carefully listed as follows.

Comment to reviewer question 4.

Question:

“Is the manuscript presented in an intelligible fashion and written in standard English?”

Answer of Reviewer #2:

“No”

Our Comment:

We conducted a major revision of the manuscript with a focus on improving the language.

Response to Comments of Reviewer #1:

Comment:

Overall, a well written manuscript. A few clarification/grammar/word chose comments.

1. Page 8, line 37. drop the "an"

2. Page 13, line 144. prior "to" use (add the "to")

3. Page 13, line 146. change "was given into" to "was placed within"

4. Page 14, line 191. change "drastically rose" to "significantly increased"

5. Page 20, line 320. This sentence seems to be missing words/text.

6. Page 21. line 355. change "these results can" to "these results it can"

7. PAge 22, line 395 change "these results can be" to "these results it can"

Response:

 1.-7. All changes were included.

Response to Comments of Reviewer #2:

Comment:

The correction points of this paper are listed below

a) The authors might choose more suitable title for their manuscript.

b) The novelty of this work isn’t clear. The authors should clarify the novelty of this work. It is better to show the novelty of this work by comparing it with the previous researches. Try to prepare a table for comparing the results.

c) Enhance the literature review of articles by using recent articles especially from journal of PLOS ONE. Most part of references are too old.

d) The abbreviation should be removed from ABSTRACT.

e) The language/writing needs improvement throughout the manuscript.

f) More explain should be presented in the result section.

a) My suggestion is that enhance the introduction section by splitting it to more paragraphs. Try to discuss the aim of the study and the novelty of the research in more details.

b) Abstract and conclusion must be improved.

c) All of figures don’t have good enough quality. The authors should correct them.

d) Keywords should be written!!

e) There is not any mention to the application of the results of this study in industry, present some application of this work.

f) The below paragraph needs some references

Page 3 line 69: “In previous studies, only a few model contaminants with similar degradation properties or 70 chemical dosimetries that are susceptible for only one type of ROS were used for investigation 71 and optimization of the pyrocatalytic process. This approach has the risk to optimize the 72 material, reaction and process parameters towards this one model contaminant or type of ROS. 73 Additionally, the influence of pH changes on the model contaminants, chemical dosimetries 74 and the pyrocatalytic process was neglected so far. Instead, we propose to use a pH-independent 75 and less specific redox assays for indirect ROS detection with the aim to optimize the 76 unselective overall oxidation capability of the pyrocatalysis. In this way, the pyrocatalytic 77 process will be applicable for the oxidative removal of a broad variety of contaminants in a 78 broad variety of matrices”

Response:

a) We changed the title to: “Pyrocatalysis – The DCF-assay as a pH-robust tool to determine the oxidation capability of thermally excited pyroelectric powders”

b) We clarified the novelty of this work, improved the introduction and included a table for comparison with previous studies.

c) We expanded the literature review of articles by using recent articles especially from journal PLOS ONE. It was still necessary to keep some older articles as references for programs (MAUD), cif-files or original methods (DCF-assay) that we used in our work or to explain some unexpected findings.

d) We removed all abbreviations from the ABSTRACT.

e) We improved the language/writing throughout the manuscript.

f) We critically reviewed the results section.

a) We improved the introduction section by splitting it to more paragraphs and discussed the aim of the study and the novelty of the research with more details.

b) We improved the abstract and the conclusion.

c) All figures meet the PLOS requirements. We checked this with the Preflight Analysis and Conversion Engine (PACE) digital diagnostic tool (https://pacev2.apexcovantage.com/). The quality of our figures was automatically decreased during the process of creating the manuscript in the submission process. When we downloaded the tif-files for example on page 31 with the link in the upper right corner the quality was good.

d) The keywords were already given and listed here again:

• water remediation; advanced oxidation process; energy harvesting; reactive oxygen species; pyroelectric effect; pyrocatalysis; DCF oxidation assay; barium titanate; lithium tantalate; lithium niobate

e) We improved the introduction section and discussed the aim of the study and the application of the pyrocatalysis as an energy harvesting water remediation technique with low energy costs.

f) We added references to the paragraph.

---

## [Decision Letter · Decision Letter 1]

22 Jan 2020

Pyrocatalysis - The DCF assay as a pH-robust tool to determine the oxidation capability of thermally excited pyroelectric powders

PONE-D-19-27422R1

Dear Dr. Braeutigam,

We are pleased to inform you that your manuscript has been judged scientifically suitable for publication and will be formally accepted for publication once it complies with all outstanding technical requirements.

With kind regards,

Dr Mason Sarafraz

PLOS ONE

Additional Editor Comments (optional):

Reviewers' comments:

Reviewer's Responses to Questions

**Comments to the Author**

1. If the authors have adequately addressed your comments raised in a previous round of review and you feel that this manuscript is now acceptable for publication, you may indicate that here to bypass the “Comments to the Author” section, enter your conflict of interest statement in the “Confidential to Editor” section, and submit your "Accept" recommendation.

Reviewer #1: All comments have been addressed

Reviewer #2: All comments have been addressed

2. Is the manuscript technically sound, and do the data support the conclusions?

Reviewer #1: Yes

Reviewer #2: Yes

3. Has the statistical analysis been performed appropriately and rigorously? 

Reviewer #1: Yes

Reviewer #2: (No Response)

4. Have the authors made all data underlying the findings in their manuscript fully available?

Reviewer #1: Yes

Reviewer #2: (No Response)

5. Is the manuscript presented in an intelligible fashion and written in standard English?

Reviewer #1: Yes

Reviewer #2: (No Response)

6. Review Comments to the Author

Reviewer #1: The authors adequately addressed the concerns raised. I have no further comments.

..

Reviewer #2: (No Response)

7. PLOS authors have the option to publish the peer review history of their article (what does this mean?). If published, this will include your full peer review and any attached files.

Reviewer #1: No

Reviewer #2: No

---

## [Editor Report · Acceptance letter]

28 Jan 2020

PONE-D-19-27422R1 

Pyrocatalysis - The DCF assay as a pH-robust tool to determine the oxidation capability of thermally excited pyroelectric powders 

Dear Dr. Braeutigam:

I am pleased to inform you that your manuscript has been deemed suitable for publication in PLOS ONE. Congratulations! Your manuscript is now with our production department. 

With kind regards,

on behalf of

Dr. Mason Sarafraz 

Academic Editor

PLOS ONE